# Centrosome age regulates kinetochore–microtubule stability and biases chromosome mis-segregation

Ivana Gasic[1,2], Purnima Nerurkar[2], Patrick Meraldi[1]*

[1]Department of Cellular Physiology and Metabolism, University of Geneva, Geneva, Switzerland; [2]Institute of Biochemistry, Eidgenössische Technische Hochschule Zürich, Zürich, Switzerland

**Abstract** The poles of the mitotic spindle contain one old and one young centrosome. In asymmetric stem cell divisions, the age of centrosomes affects their behaviour and their probability to remain in the stem cell. In contrast, in symmetric divisions, old and young centrosomes are thought to behave equally. This hypothesis is, however, untested. In this study, we show in symmetrically dividing human cells that kinetochore–microtubules associated to old centrosomes are more stable than those associated to young centrosomes, and that this difference favours the accumulation of premature end-on attachments that delay the alignment of polar chromosomes at old centrosomes. This differential microtubule stability depends on cenexin, a protein enriched on old centrosomes. It persists throughout mitosis, biasing chromosome segregation in anaphase by causing daughter cells with old centrosomes to retain non-disjoint chromosomes 85% of the time. We conclude that centrosome age imposes via cenexin a functional asymmetry on all mitotic spindles.

*For correspondence: Patrick.
meraldi@unige.ch

Competing interests: The authors declare that no competing interests exist.

## Introduction

The bipolar spindle has a symmetric appearance; nevertheless it contains two centrosomes of different ages, as every centrosome is duplicated once during the cell cycle, resulting in the presence of an old and young centrosome at mitotic onset (*Nigg and Stearns, 2011*). In asymmetric stem cell divisions centrosome age differentially affects their capacity to nucleate microtubules and their positioning with respect to the polarity and cell division axis (*Yamashita et al., 2007*; *Wang et al., 2009*; *Januschke et al., 2013*). Stem cells stereotypically inherit the centrosome, which nucleates more microtubules, which in most cases is the old centrosome (except in fly neuroblast divisions, where stem cells retain the young active centrosome; *Januschke et al., 2011, Conduit and Raff, 2010*). Old centrosomes also co-segregate with the ciliary membrane in stem cell divisions, allowing daughter stem cells to form a primary cilium earlier than the differentiating daughter cells (*Paridaen et al., 2013*). In symmetric divisions, the old and young centrosomes can be differentiated at the ultra-structural level, and in terms of their microtubule-anchoring capacity during interphase (*Rieder and Borisy, 1982*; *Piel et al., 2000*). The oldest centriole within the old centrosome contains distal and subdistal appendages: the first are necessary for centrioles to become basal bodies that can contact the plasma membrane (*Graser et al., 2007*; *Hoyer-Fender, 2010*), while the latter are involved in the organization of the interphase microtubule network, due to the presence of ninein, a key microtubule-anchoring protein (*Mogensen et al., 2000*). Both structures require the presence of cenexin, the oldest known marker for old centrosomes and appendages (*Lange and Gull, 1995*; *Ishikawa et al., 2005*). Importantly, all these structural differences disappear progressively as cells enter mitosis; therefore, it is assumed that the old and the young centrosomes behave indistinguishably in symmetrically dividing cells, resulting in a symmetric bipolar spindle. This hypothesis has, however,

**eLife digest** Cells are able to copy their DNA and then divide to make two daughter cells that each have a complete set of genetic material. In animal cells, the DNA is arranged within structures called chromosomes and groups of proteins called centrosomes control the process that separates the chromosome copies as the cell divides.

Each cell starts off with one centrosome, but before it divides, this centrosome is copied so that the cell now has two centrosomes at opposite ends of the cell, one old and one new. Filaments called microtubules assemble from the centrosomes and attach to the chromosomes. The microtubules first align all the chromosomes in the middle of the cell before pulling them towards the centrosomes as the cell divides.

Some cells divide such that the two daughter cells are destined to take on different roles, for example, a stem cell may divide to produce one stem cell and one skin cell. The end of the dividing cell that will become the stem cell contains the older centrosome, while the half that forms the skin cell will receive the younger centrosome. Other cells in the body may divide to form daughter cells that have the same fate, known as symmetrical division. In these cases, it is thought that there is no difference between the behaviour of the old and young centrosomes, but this idea has never been tested.

Here, Gasic et al. studied symmetrical division of human cells using fluorescent tags that made it possible to tell the centrosomes apart. The experiments show that the microtubules that assemble from the older centrosome bind the chromosome more tightly than those that form from the younger centrosome. This delays the alignment of the chromosomes that are connected to the old centrosome, as this process requires a flexible attachment. Moreover, in case the two chromosome copies fail to separate properly as cells divide, the older centrosome is more likely to receive both chromosome copies at the expense of the other centrosome. A protein called cenexin is present at higher levels around older centrosomes than around younger ones and is responsible for this effect.

Gasic et al.'s findings show that the age of the centrosomes leads to asymmetry in all cell divisions, even those that produce cells that are destined to have the same role in an organism. The next challenge will be to understand whether this asymmetry has any consequences for cells, in particular cancer cells.

never been directly tested at the functional level. Here, we tested whether centrosome age affects cell division in symmetrically dividing human cells, focusing on the ability of centrosomes to organize the alignment and segregation of sister-chromatids into two daughter cells.

## Results

The first key task of the mitotic spindle is to bind to chromosomes via kinetochores and align them onto the metaphase plate (*Kops et al., 2010*). To distinguish between old and young centrosomes, we used untransformed hTert-RPE1 and transformed HeLa cell lines expressing eGFP-centrin1, a centriolar protein whose abundance correlates with centriole age, or an anti-cenexin antibody, a marker for old centrosomes (*Figure 1A,B*, *Kuo et al., 2011*; *Lange and Gull, 1995*). In the vast majority of the cases both markers were enriched at the same centriole pair, indicating a robust recognition of the old centrosomes (data not shown). To investigate whether half-spindles associated with old or new centrosomes align chromosomes with the same efficiency, we analyzed late prometaphase cells that contained few unaligned chromosomes. We found that 61.23% of the unaligned chromosomes were in the vicinity of old centrosomes in Hela-eGFP-centrin1 cells as opposed to 50% expected for an unbiased distribution, suggesting a difference in the efficiency of chromosome alignment (*Figure 1C,D*, statistical tests for the chromosome alignment assays throughout the study are shown in *Table 1*). As such unaligned chromosomes were rare, we also treated cells with 10 ng/ml nocodazole, a condition that moderately stabilizes microtubules (*Vasquez et al., 1997*), and that delays chromosome alignment, leading to 3–6 unaligned chromosomes per cell (*Figure 1C*). Unaligned chromosomes were again preferentially found in the vicinity of the old centrosomes in HeLa eGFP-centrin1 (63.9%) and hTert-RPE1-eGFP-centrin1 cells (71.8%), confirming the bias in chromosome alignment (*Figure 1E*). We found the same bias (67.8%) in wild-type nocodazole-treated hTert-RPE1 cells stained with cenexin, excluding

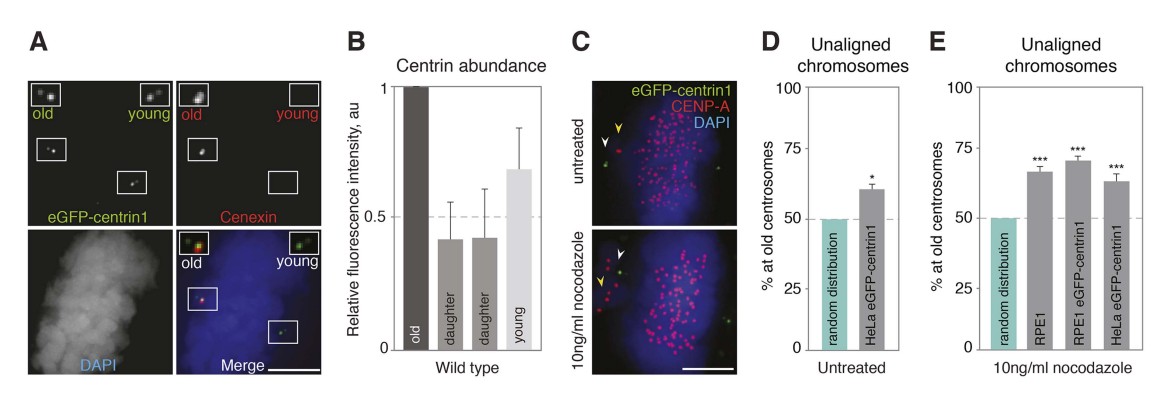

**Figure 1**. Centrosome age affects chromosome alignment. (**A**) HeLa-eGFP-centrin1 (green) cell stained for cenexin (red, old centrosome marker) and DAPI (blue, DNA). One spindle pole contains the old centriole (brightest centrin1 signal and cenexin positive) and an accompanying daughter centriole (dim signal), which together form the old centrosome. The other spindle pole contains the young centriole (intermediate centrin1 signal), which is also accompanied by a daughter centriole and which together form the young centrosome. Scale bar in all panels = 5 μm. (**B**) Amounts of eGFP-centrin1 on the old, young and daughter centrioles in HeLa-eGFP-centrin1 cells determined from 3 independent experiments in 140 cells. (**C**) Untreated HeLa-eGFP-centrin1 cell (upper panel) and hTert RPE-eGFP-centrin1 cell treated with 10 ng/ml nocodazole (lower panel) stained for CENP-A (kinetochore marker) and DAPI. Yellow arrowheads indicate unaligned chromosomes; white arrowheads old centrosomes. (**D** and **E**) Proportion of unaligned chromosomes at old centrosomes in HeLa-eGFP-centrin1 cells (**D**), and in RPE1 cells stained for cenexin, RPE1-eGFP-centrin1 cells and HeLa-eGFP-centrin1 cells treated with 10 ng/ml nocodazole (**E**). For experiment and cell numbers, and p-values see *Table 1*. For results of individual experiments see *Figure 1—source data 1*. Error bars indicate s.e.m. * indicates $p \leq 0.05$ in Binomial test compared to random distribution, *** indicates $p \leq 0.01$ in Binomial test compared to random distribution.

The following source data is available for figure 1:

**Source data 1**. Values of individual experiments of graphs shown in *Figure 1*.

any effect due to eGFP-centrin1 expression (*Figure 1E*). We conclude that the half-spindles associated to the old centrosomes accumulate more unaligned chromosomes or that unaligned chromosomes align less efficiently when bound to microtubules emanating from the old centrosomes.

The bias in unaligned chromosomes could reflect faster kinetics in the initial capture of sister-kinetochore pairs by old centrosomes, for example, because they are closer to kinetochores at nuclear envelope breakdown or because they mature—that is, acquire a high, mitotic microtubule-nucleating capacity—earlier. Alternatively, the bias could reflect a permanent difference between the two centrosomes to capture or to align chromosomes. To test whether at mitotic onset old centrosomes capture kinetochores faster because they are closer, we compared the distances between kinetochores and old and young centrosomes at mitotic onset, but found no difference (*Figure 2A*). We next forced hTert-RPE1 or HeLa cells to enter mitosis with monopolar spindles by treating them with monastrol, a reversible inhibitor of Eg5, the kinesin that separates centrosomes (*Mayer et al., 1999*). A monastrol washout led to bipolar spindles with few unaligned chromosomes, 74.4% of which were adjacent to old centrosomes, indicating that the bias is independent of the initial centrosome position (*Figure 2B,C*; and *Figure 2—figure supplement 1*). To test whether old centrosomes capture more kinetochores because they mature earlier, we treated cells with high doses of nocodazole (1 μg/ml), allowing them to enter mitosis without microtubules and to fully mature the two centrosomes (*Khodjakov and Rieder, 1999*), before washing out nocodazole for 1 hr: 62.6% of the unaligned chromosomes were adjacent to old centrosomes, indicating that the alignment bias reflects a permanent difference between the centrosomes that is independent of the initial conditions at mitotic onset (*Figure 2B,C*). The two washout experiments also confirmed that this bias does not require low nocodazole concentrations, since in both cases, cells were released in nocodazole-free medium.

In asymmetric cell divisions the old and young centrosomes have different capacities to nucleate microtubules, providing a key clue for centrosome positioning and inheritance (*Wang et al., 2009*; *Januschke et al., 2013*; *Lerit and Rusan, 2013*). If microtubule nucleation from the two centrosomes also differed in symmetric cell division, this might allow one centrosome to capture more kinetochores.

**Table 1**. Percentage of unaligned chromosomes at old centrosomes

| Condition | N° of repeats | N° of cells | N° of chromosomes | % Chromosomes at old centrosomes | 2-tailed Binomial test p |
|---|---|---|---|---|---|
| HeLa-eGFP-centrin1 DMSO | 3 | 33 | 93 | 61.23 | 0.037 |
| hTert-RPE1 10 ng/ml nocodazole | 3 | 161 | 227 | 67.80 | 8.08e-8 |
| hTert-RPE1-eGFP-centrin1 10 ng/ml nocodazole | 7 | 127 | 295 | 71.81 | 1.0e-12 |
| HeLa-eGFP-centrin1 10 ng/ml nocodazole | 3 | 57 | 532 | 63.91 | 1.42e-10 |
| hTert-RPE1 Eg5 inhibition recovery | 3 | 53 | 156 | 74.36 | 8.9e-10 |
| hTert-RPE1 nocodazole recovery | 3 | 59 | 164 | 61.58 | 0.00373 |
| HeLa-eGFP-CENP-A/eGFP-centrin1 10 ng/ml nocodazole | 3 | 142 | 946 | 58.03 | 8.68e-07 |
| HeLa-eGFP-CENP-A/eGFP-centrin1 Eg5 inhibition recovery | 5 | 68 | 306 | 61.11 | 0.000121 |
| hTert-RPE1-eGFP-centrin1 siCtrl 10 ng/ml nocodazole | 3 | 92 | 206 | 68.45 | 1.26e-07 |
| hTert-RPE1-eGFP-centrin1 siNinein 10 ng/ml nocodazole | 4 | 77 | 169 | 66.86 | 0.0000138 |
| hTert-RPE1-eGFP-centrin1 CENP-E inhibitor | 3 | 65 | 393 | 49.87 | n.s* |
| hTert-RPE1-eGFP-centrin1 CENP-E inhibitor 10 ng/ml nocodazole | 3 | 59 | 459 | 52.29 | n.s* |
| hTert RPE-eGFP-centrin1 5 nM Taxol | 3 | 50 | 105 | 43.81 | n.s* |
| hTert RPE-eGFP-centrin1 siDsn1 10 ng/ml nocodazole | 3 | 42 | 162 | 43.82 | n.s* |
| hTert RPE-eGFP-centrin1 siNnf1 10 ng/ml nocodazole | 5 | 21 | 46 | 52.17 | n.s* |

*Non-significant.

However, a microtubule re-nucleation assay revealed no difference in microtubule nucleation capacity between the two centrosomes in HeLa and RPE1 cells (*Figure 2D,E*), suggesting that the centrosomal microtubule nucleation capacity did not cause the biased distribution of unaligned chromosomes. To study if chromosome alignment process per se is asymmetric, we inhibited the Centromere-associated Protein E (CENP-E), the kinetochore-bound kinesin that aligns polar chromosomes by transporting them along existing spindle microtubules (*Wood et al., 1997*; *Kapoor et al., 2006*; *Barisic et al., 2014*). Partial CENP-E inhibition, yielding few polar chromosomes, abolished the bias in the distribution of unaligned chromosomes in the absence or presence of 10 ng/ml nocodazole, (*Figure 2F, G*, 49.87% and 52.29% respectively), indicating that the bias depends on CENP-E, and that chromosome alignment itself is biased by centrosome age.

The CENP-E-dependent alignment bias could be due to an asymmetric abundance of CENP-E; however, the levels of CENP-E on unaligned chromosomes associated to old or young centrosomes were equal (*Figure 3A,B* and *Figure 3—figure supplement 1A*). Alternatively, since CENP-E favours lateral kinetochore–microtubule attachments to transport unaligned chromosomes towards the metaphase plate (*Kapoor et al., 2006*), we reasoned that a difference in the types of kinetochore–microtubule attachments might bias the alignment of unaligned chromosomes: specifically end-on attachments might delay CENP-E driven chromosome alignment, by creating a poleward drag. Indeed, chromosomes that are not captured by microtubules emanating from both poles, bind laterally to microtubules from the closest pole, and are first driven to this pole in a dynein-dependent manner, before CENP-E aligns them on the metaphase plate (*Barisic et al., 2014*). During these movements, kinetochores can in some cases form end-on monotelic or syntelic attachments. These non-bipolar end-on attachments are normally destabilized in an Aurora-B-dependent manner (*Hauf et al., 2003*), favouring the formation of lateral

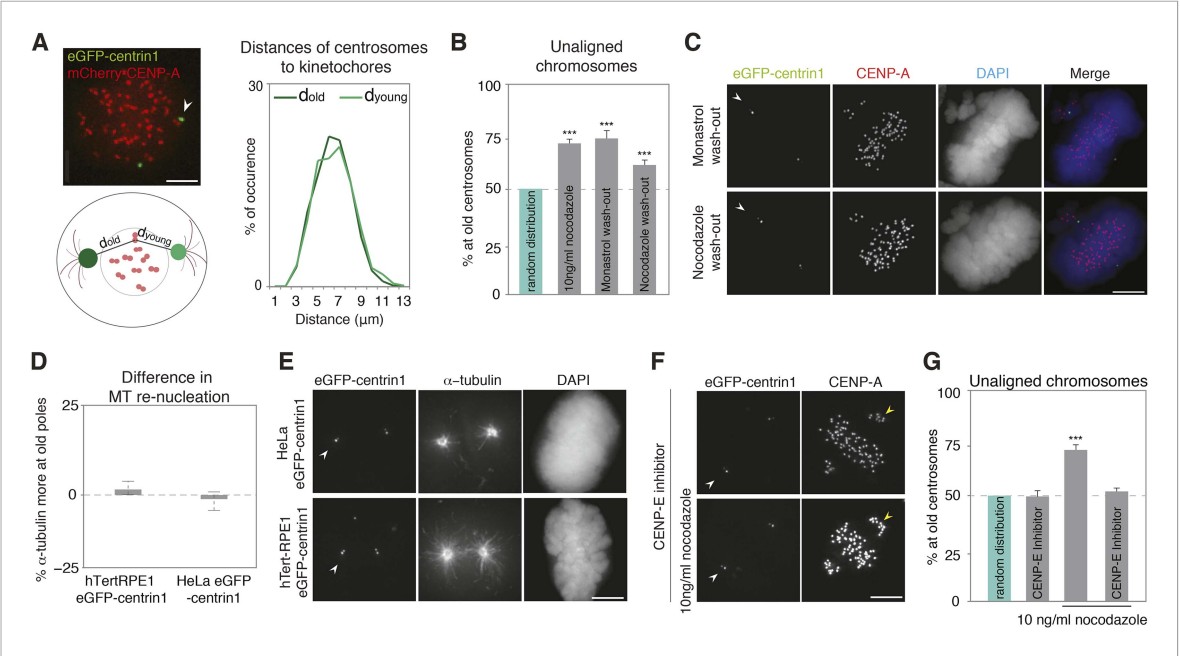

**Figure 2**. The asymmetric distribution of unaligned chromosomes depends on CENP-E. (**A**) An illustrative example of a HeLa-eGFP-centrin1/mCherry-CENP-A cell where the distances from centrosomes to kinetochores were measured 30 s before nuclear envelope breakdown (left top), assay to calculate the distances between kinetochores and centrosomes (left bottom), and distribution (right) of the measured distances. Values are determined from 24 cells and 1434 kinetochores in 6 independent experiments. White arrowheads indicate old centrosomes in all panels. Scale bars in all panels = 5 µm. (**B**) Proportion of unaligned chromosomes at the old centrosomes in hTert-RPE1-eGFP-centrin1 cells after indicated treatments. Error bars indicate s.e.m. *** indicates $p \leq 0.01$ in Binomial test. (**C**) hTert-RPE1-eGFP-centrin1 cells stained for CENP-A and DAPI after indicated treatments. (**D**) Differences in the intensity of the microtubule asters in a re-nucleation assay at old and young centrosomes as shown in E, calculated in 32–49 cells in 3 independent experiments. Columns indicate the median, errors bars the 99% CI. Precise methodology is shown in *Figure 2—figure supplement 2*. (**E**) HeLa-eGFP-centrin1 or hTert-RPE1-eGFP-centrin1 cells stained for α-tubulin after a microtubule re-nucleation assay. (**F**) hTert-RPE1-eGFP-centrin1 cells treated with 10 ng/ml nocodazole and/or CENP-E inhibitor, and stained for CENP-A. Yellow arrowheads indicate unaligned kinetochores in the proximity of the young centrosome (**G**) Proportion of unaligned chromosomes at old centrosomes in hTert-RPE1-eGFP-centrin1 cells treated with 10 ng/ml nocodazole and/or CENP-E inhibitor. Error bars indicate s.e.m. *** indicates $p \leq 0.01$ in Binomial test. For results of all individual experiments see *Figure 2—source data 1*.

The following source data and figure supplements are available for figure 2:

**Source data 1**. Values of individual experiments of graphs shown in *Figure 2*.

**Figure supplement 1**. Monastrol wash-out does not change the alignment bias in HeLa cells.

**Figure supplement 2**. Methodology to compare microtubule re-nucleation at old and new pole.

kinetochore–microtubule attachments. However, if end-on attachments were to be more stable at one centrosome, they would delay this conversion and create a drag on the CENP-E driven alignment. To test this hypothesis, we visualized by 3D-high-resolution microscopy kinetochore–microtubule attachments of individual, single kinetochores on unaligned chromosomes in cells treated with 10 ng/ml nocodazole and fixed with glutaraldehyde. At old centrosomes nearly three times more individual kinetochores had end-on attachments (13.0% vs 4.7% at the young centrosome, $p = 0.003$ in paired t-test) and fewer lateral attachments (83.6% vs 89.1% at the young centrosomes; $p = 0.0007$ in paired t-test; overall $p = 0.0024$ in 2way-ANOVA-test; *Figure 3C,D* and *Figure 3—figure supplement 2*); the number of unattached kinetochores was higher at young centrosomes, even tough this difference was statistically not significant ($p = 0.06$ in paired t-test; *Figure 3—figure supplement 2*). This implied an overall higher stability of kinetochore–microtubules at old centrosomes. To confirm this result, we quantified the levels of tubulin acetylation on individual kinetochore–fibres of sister-kinetochore pairs

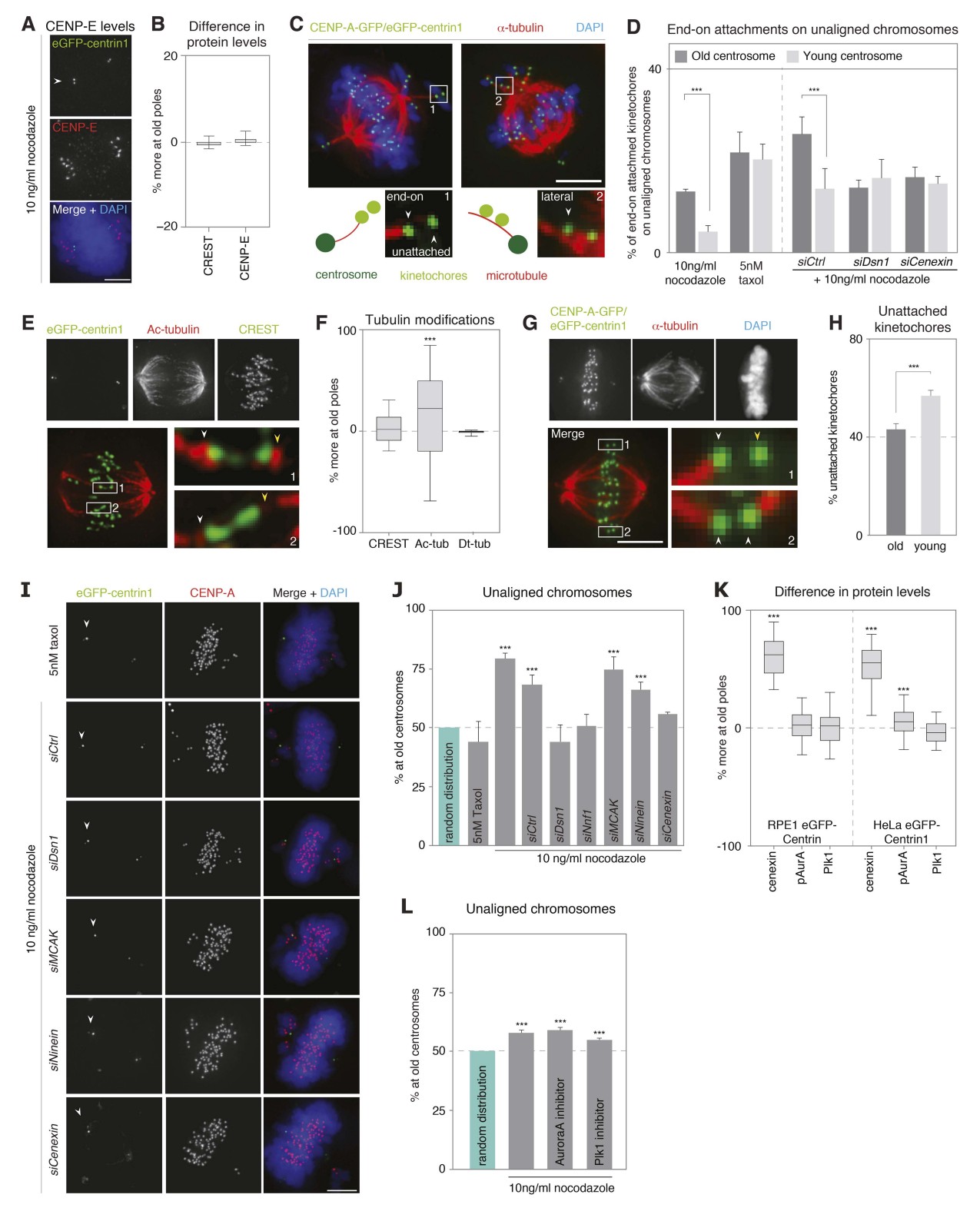

**Figure 3**. Kinetochore–microtubules bound to old centrosomes are more stable. (**A**) hTert-RPE1-eGFP-centrin1 cells treated with 10 ng/ml nocodazole and stained for CENP-E and DAPI. White arrowheads indicate old centrosomes in all panels. Scale bars in all panels = 5 μm. (**B**) Differences in the abundance of CENP-E and CREST (kinetochore marker) at kinetochores bound to old and young centrosomes, calculated from 27 cells in 3 independent experiments. Columns indicate the median; error bars the 99% CI. (**C**) Immunofluorescence image of a HeLa-eGFP-centrin1/eGFP-CENP-A (green) cells

*Figure 3. continued on next page*

*Figure 3. Continued*

treated with 10 ng/ml nocodazole, fixed with glutaraldehyde, and stained for α-tubulin (red) and DAPI (blue). Single kinetochores in every unaligned sister-kinetochore pair were classified as end-on attached, laterally attached or unattached. Inset 1 on the left shows an illustrative example of a kinetochore pair with one unattached and one end-on attached kinetochore; inset 2 on the right shows an illustrative example with 2 laterally attached kinetochores. (**D**) Quantification of individual end-on attached kinetochores at old and young centrosomes in HeLa-eGFP-centrin1/eGFP-CENP-A cells treated with 10 ng/ml nocodazole, 5 nM taxol and the indicated siRNAs and 10 ng/ml nocodazole. Percentages are based on 3 independent experiments with 29–50 cells. Error bars indicate s.e.m; *** indicates $p \leq 0.01$ in paired t-test. (**E**) hTertRPE1-eGFP-centrin1 cells stained with anti-acetylated tubulin (red) and CREST (green) antibodies. Shown are total projections (upper panels) or maximum-intensity projections of 5–10 planes around the focal plane of interest (lower panels). White arrowheads indicate kinetochore–microtubules with stronger acetylation, yellow with weaker acetylation. Note that the white arrows are on the side of the old centrosome. (**F**) Differences in the abundance of acetylated tubulin on k-fibres of sister-kinetochores, and detyrosinated tubulin on the two spindle halves in hTertRPE1-eGFP-centrin1 cells, based on 3 independent experiments and 32–33 cells. Methodology is explained in *Figure 3—figure supplement 2*. Columns indicate the meadian, error bars the 99% CI. (**G**) HeLa-eGFP-centrin1/eGFP-CENP-A cells treated with 0.5 mM $Ca^{2+}$ for 10 min stained for α-tubulin (red) and DAPI (blue). Shown are total projections (upper panels) or maximum-intensity projections of 5–10 planes around the focal plane of interest (lower panels). White arrowheads in zoom-ins indicate end-on attached kinetochores and yellow arrow the unattached kinetochore. (**H**) Percentage of unattached kinetochores oriented towards old or young poles based on 3 independent experiments and 33 cells. (**I**) hTert-RPE1-eGFP-centrin1 cells stained with CENP-A antibodies (red) and DAPI (blue) after treatment with 5 nM taxol or the indicated siRNAs and 10 ng/ml nocodazole. White arrowheads indicate old centrosome. (**J**) Proportion of unaligned chromosomes at old centrosome in hTert-RPE1-eGFP-centrin1 cells treated with 5 nM taxol or with 10 ng/ml nocodazole after the indicated siRNA treatment. Error bars indicate s.e.m; *** indicates $p \leq 0.01$ in Binomial test. (**K**) Differences in the abundance of cenexin, phospho-Aurora-A, and Plk1 at old and young centrosomes in HeLa and hTert-RPE1-eGFP-centrin1 cells, based on 3 independent experiments and 41–113 cells. Methodology is explained in *Figure 3—figure supplement 2*. Columns indicate the median, error bars the 99% CI. (**L**) Proportion of unaligned chromosomes at old centrosome in HeLa-eGFP-centrin1 cells treated with Aurora-A or Plk1 inhibitors. Error bars indicate s.e.m; *** indicates $p \leq 0.01$ in Binomial test. For results of all individual experiments see *Figure 3—source data 1*.

The following source data and figure supplements are available for figure 3:

**Source data 1**. Values of individual experiments of graphs shown in *Figure 3*.

**Figure supplement 1**. Methodology to compare kinetochore- and centrosome-associated protein intensities at old and young spindle poles.

**Figure supplement 2**. Quantification of the proportion of laterally and unattached kinetochores.

**Figure supplement 3**. Quantification of kinetochore–microtubule stability in cold treated cells.

**Figure supplement 4**. Validation of siRNA treatments.

---

aligned on the metaphase plate, as tubulin acetylation preferentially accumulates on stable microtubules (*Webster and Borisy, 1989*). This analysis revealed higher level of acetylation on kinetochore–microtubules associated with old centrosomes (median difference of 22% in tubulin acetylation at the plus ends of microtubules attached to sister-kinetochores vs 2% in CREST levels between sister-kinetochores, $p < 0.0001$ in Wilcoxon Signed Rank Test; *Figure 3E,F* and *Figure 3—figure supplement 1C*). In contrast, when we measured the levels of detyrosinated tubulin, a modification that has been linked to preferential CENP-E motor activity (*Barisic et al., 2015*), we found no difference (median difference of 0.3%; *Figure 3F* and *Figure 3—figure supplement 1D*). To also functionally confirm the difference in microtubule stability, metaphase cells were treated for 10 min with 0.5 mM $Ca^{2+}$, a condition that gradually destabilizes microtubules, before fixing them with glutaraldehyde and staining for kinetochores and microtubules. While such a treatment did not reveal strong overall differences in the two half-spindles (*Figure 3G*), a detailed analysis of kinetochore–microtubule attachments of aligned sister-kinetochores revealed that kinetochores oriented towards young centrosomes were significantly more likely to have lost their attachment, than those oriented towards the old centrosome (56.8% vs 43.2%; $p = 0.014$ in paired t-test; *Figure 3G,H*). Together, these data indicated that the kinetochore–microtubules emanating from old centrosomes are more stable.

To test if the difference in kinetochore–microtubule stability is at the origin of the alignment bias, we depleted the kinetochore proteins Dsn1 or Nnf1 (both Mis12 complex), or treated the cells with a low dose of taxol (5 nM). These conditions strongly destabilize kinetochore–microtubules (Dsn1 and Nnf1 depletion; (*Kline et al., 2006*), or hyperstabilize kinetochore–microtubules (taxol; *Figure 3—figure supplement 3*). Either treatment abolished the bias in alignment and equalized the number of end-on

attached unaligned kinetochores (*Figure 3D,I,J*). In contrast, knock-down of the Mitotic Centromere-Associated Kinesin (MCAK), a microtubule depolymerase that is required for destabilization of erroneous kinetochore–microtubule attachments (*Knowlton et al., 2006*), whose depletion leads to mild microtubule stabilization in metaphase, but not in prometaphase ((*Bakhoum et al., 2009*) and *Figure 3—figure supplement 3*), did not change the bias in chromosome alignment (*Figure 3I,J*). This suggested that a massive stabilization or destabilization of all kinetochore–microtubules equilibrates the difference in kinetochore–microtubule stability and chromosome alignment, but that a mild stabilization does not change this bias. We conclude that the difference in kinetochore–microtubule stability biases chromosome alignment.

Which factors at centrosomes could generate an age-dependent difference in kinetochore–microtubule stability causing a bias in chromosome alignment? We first considered two centrosomal kinases, Aurora-A and Plk1, which can both affect kinetochore–microtubule stability (*Liu et al., 2012*; *Bakhoum et al., 2014*). We compared by quantitative immunofluorescence the levels of Plk1 or the activity of Aurora-A (with an antibody that is specific for active Aurora-A) at old and new centrosomes, to reveal a potential asymmetry in kinase levels/activity. While Plk1 was symmetrically distributed, we found a modest increase of active Aurora-A on old centrosomes in HeLa cells (*Figure 3K* and *Figure 3—figure supplement 1B*). This difference was, however, not present in RPE1 cells (*Figure 3K*); moreover inhibition of Aurora-A or Plk1 did not abolish the bias in chromosome alignment, indicating that it does not depend on these two kinases (*Figure 3L*). In a second step, we investigated the possible involvement of ninein, as it is essential for cell fate determination in asymmetric cell divisions of neuronal progenitors and preferentially localizes to old centrosomes in asymmetric cell division (*Wang et al., 2009*), and of cenexin itself, the classical marker for old centrosomes (*Figure 3K*—note that ninein levels could not be compared on old and young mitotic centrosomes, as it is only present at very low levels (*Logarinho et al., 2012*)). While ninein depletion had no effect on chromosome alignment, cenexin depletion randomized the distribution of unaligned chromosomes (*Figure 3I,J*). Furthermore, it also equalized the percentage of end-on attached kinetochores at unaligned chromosomes, indicating that cenexin affects kinetochore–microtubule stability (*Figure 3D*). We conclude that old centrosomes stabilize kinetochore–microtubules in a cenexin-dependent manner.

The ultimate function of the mitotic spindle is to accurately segregate sister chromatids. If kinetochore–microtubules bound to the old centrosomes were more stable, we predicted that this should affect the fate of chromosomes that fail to fully disjoin in anaphase; chromosome non-disjunction is a frequent cause of chromosome mis-segregation in cancer cells, that can be caused by various defects, such as stretches of unreplicated DNA, telomere fusions, or chromosome entanglements (*Aguilera and García-Muse, 2013*). To monitor the fate of such chromosome non-disjunction, we monitored by live-cell imaging HeLa-eGFP-centrin1/mCherry-CENP-A cells released for synchronization purpose from a monastrol arrest. As previously reported, this procedure produced a number of single lagging, most likely merotelic, kinetochores, whose exact fate could not be tracked. However, in addition in roughly 2–5% of anaphases, we observed the presence of two lagging kinetochores that moved in synchrony between the two daughter DNA masses, but were separated by several microns, suggesting a sister-kinetochore pair on a non-disjoint chromosome (*Figure 4A* and *Video 1*). This assumption was confirmed by high-resolution immunofluorescence imaging, as such kinetochore pairs were invariably connected by a DNA thread (see representative images in *Figure 4B*). In those instances where both sister-kinetochores segregated to the same daughter cell, we found a strong bias in chromosome mis-segregation as 18 out of the 21 analyzed kinetochore pairs co-segregated with the old centrosome (*Figure 4C*; *Video 1*; number of experiments, cells and statistical tests for all chromosome mis-segregation events are in *Table 2*). This suggested that non-disjoint chromosomes are preferentially pulled towards the old centrosomes, possibly due to a higher stability of the kinetochore–microtubules emanating from the old centrosomes, which in a tug-of-war would favour a destabilization and release of the kinetochore–microtubules bound to the young pole. To test this hypothesis, we treated cells with Nnf1 and Cenexin siRNAs, which had abolished the bias in chromosome alignment and the asymmetry in the percentage of end-on attached kinetochores. In both cases, the bias in chromosome mis-segregation was abolished (*Figure 4C*; *Video 2 and 3*); in contrast when we depleted MCAK, which did not abolish the bias in chromosome alignment, chromosome mis-segregation was still biased (*Figure 4C*). We conclude that the difference in the stability of kinetochore–microtubules bound to the old or the young centrosome persists in anaphase, and that this difference causes non-disjoint chromosomes to co-segregate with old centrosomes.

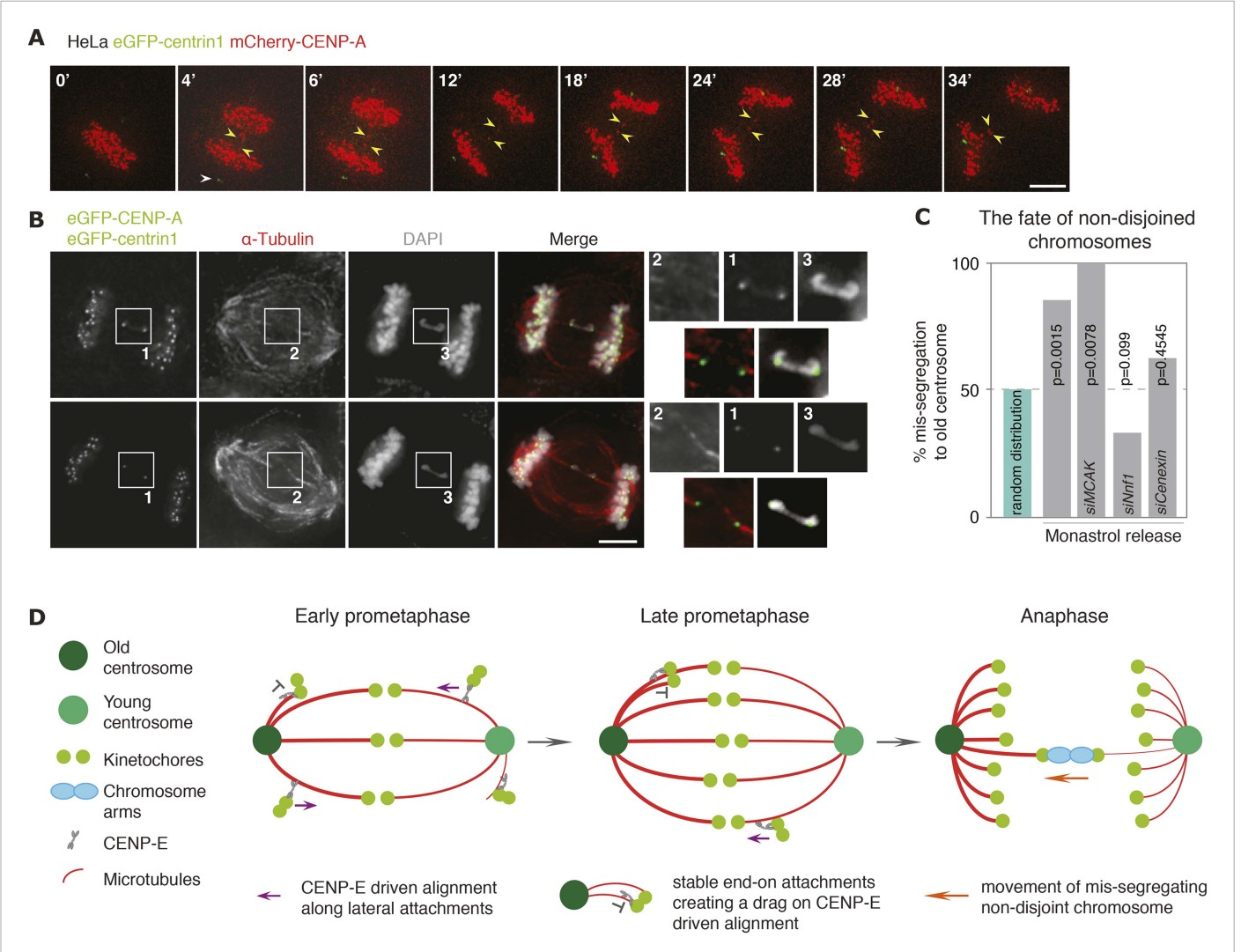

**Figure 4**. Non-disjoined chromosomes co-segregate with old centrosomes. (**A**) Time lapse images of a HeLa-eGFP-centrin1/mCherry-CENP-A cell with a non-disjoined sister-kinetochore pair in anaphase. White arrowhead indicates the old centrosome, yellow arrowheads the non-disjoined sister-kinetochore pair. Scale bar = 10 μm. (**B**) Illustrative example of a HeLa-eGFP-centrin1/eGFP-CENP-A cell in anaphase stained for α-tubulin with a non-disjoined chromosome. Insets highlight the non-disjoined chromosomes. Scale bar = 5 μm. (**C**) Proportion of non-disjoined chromosomes that co-segregate with the old centrosomes in HeLa-eGFP-centrin1/mCherry-CENP-A cells treated with the indicated siRNAs. For statistics and number of experiments, see *Table 2*. (**D**) Proposed model of how old and new centrosomes differentially affect chromosome alignment and chromosome segregation via kinetochore–microtubule stability.

## Discussion

Our results demonstrate that even in symmetrically dividing cells the two half-spindles behave in an asymmetric manner, and that centrosome age imposes a functional asymmetry on all mitotic spindles (see model *Figure 4D*). We find that this asymmetry reflects a differential stability of kinetochore–microtubules that depends on the presence of cenexin at old centrosomes, indicating that its presence influences the relative stability of kinetochore–microtubules. Previous studies demonstrated that knock-out of cenexin does not impair mitotic progression; it contributes, however, to the stability of the centrosome-bound microtubules during interphase, consistent with our findings that cenexin affects mitotic microtubules (*Ishikawa et al., 2005*; *Tateishi et al., 2013*). Cenexin is known since a long time as a marker for old centrioles (*Lange and Gull, 1995*), yet the molecular mechanisms by which it affects microtubules are unclear. It is required for the formation of distal and sub-distal appendages on the oldest centriole, two structures that are essential for the

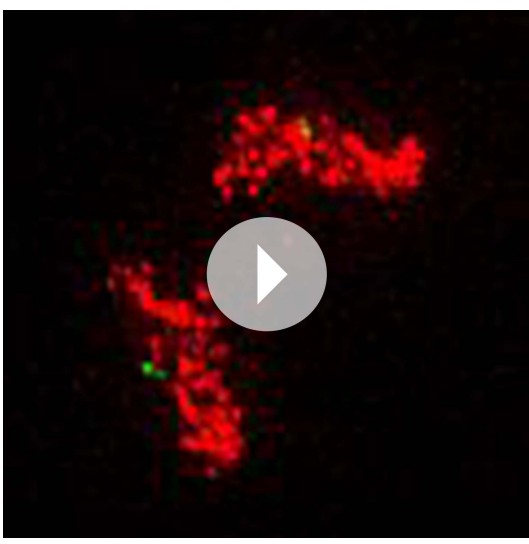

**Video 1.** HeLa cell expressing mCherry-CENP-A (kinetochore marker in red) and eGFP-centrin1 (centrosome age marker in green) after a monastrol washout. Note that the mis-segregating chromosome moves towards the brighter, old centrosome.

formation of basal bodies during ciliogenesis (*Ishikawa et al., 2005*). These structures consist of more then ten different centrosomal proteins, including CEP164, CEP123, CEP83, SCLT1 and FBF1 at the distal appendages and ninein, centriolin, ε-tubulin, trichoplein and CEP170 at sub-distal appendages (*Mogensen et al., 2000*; *Chang et al., 2003*; *Gromley et al., 2003*; *Guarguaglini et al., 2005*; *Graser et al., 2007*; *Ibi et al., 2011*; *Sillibourne et al., 2013*; *Tanos et al., 2013*; *Tateishi et al., 2013*). Moreover Plk1 has been shown to bind Odf2 at centrosomes (*Soung et al., 2009*). Future work will thus have to evaluate whether the ability to stabilize microtubules is a direct function of cenexin, or a more general function of centriolar appendages. We speculate that cenexin or some of its associated centrosomal proteins might control microtubule stability via three potential mechanisms: first, they could directly interact with microtubule plus-ends, as has been seen for γ-tubulin (*Bouissou et al., 2009*); second, they could affect microtubule plus-end dynamics by controlling the dynamics of the minus-end, a type of regulation that has been seen in the context of poleward microtubule flux (*Maddox et al., 2003*; *Ganem et al., 2005*; *Matos et al., 2009*); third they could act via centrosomal protein kinases, as it has been recently shown as a proof-of-principle that centrosomal-bound Aurora-A has the ability to regulate kinetochore–microtubule attachments (*Chmátal et al., 2015*; *Ye et al., 2015*).

The asymmetry in kinetochore–microtubule stability persists throughout mitosis and directs the fate of non-disjoint chromosomes, which co-segregate mostly with the old centrosome. At present it is unclear whether this asymmetry serves a direct purpose during mitosis, or if it a consequence of a centrosome asymmetry that is required for a non-mitotic function, but which cells have to deal with during each cell division. Possible non-mitotic purposes of this asymmetry include the necessity to only have one centriole capable of generating the basal body of a cilium, or the requirement to only have one centriole capable of anchoring interphase microtubules (*Piel et al., 2000*; *Nigg and Raff, 2009*). A possible mitotic function for an asymmetric behaviour of centrosomes is that it might protect stem cells that inherit the old centrosome, from losing non-disjoint chromosomes. This would provide a selective advantage, as haplo-insufficiency is much more frequent than triplo-lethality at the level of single genes in animal cells (*Lindsley et al., 1972*; *Torres et al., 2007*). We speculate that the asymmetric distribution of non-disjoint chromosomes might, however, also favour the rapid acquisition of new traits by co-occurrence of chromosome gains in

**Table 2**. Percentage of mis-segregating chromosomes that co-segregate with the old centrosomes

| Condition | N⁰ of repeats | N° of cells | N° of chromosomes | N° of chromosomes to the old centrosome | 2-tailed binomial test p |
|---|---|---|---|---|---|
| HeLa-eGFP-centrin1/ mCherry-CENP-A | 11 | 21 | 21 | 18 | 0.0015 |
| HeLa-eGFP-centrin1/ mCherry-CENP-A *siMCAK* | 3 | 8 | 8 | 8 | 0.0078 |
| HeLa-eGFP-centrin1/ mCherry-CENP-A *siNnf1* | 6 | 30 | 30 | 10 | 0.0990 |
| HeLa-eGFP-centrin1/ mCherry-CENP-A *siCenexin* | 6 | 16 | 16 | 10 | 0.4545 |

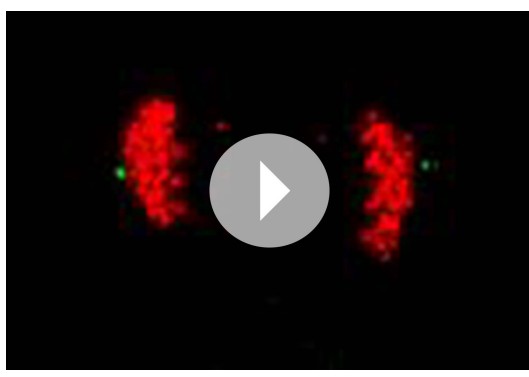

**Video 2.** HeLa cell expressing mCherry-CENP-A (kinetochore marker in red) and eGFP-centrin1 (centrosome age marker in green) depleted of Nnf1, after a monastrol washout. Note that the mis-segregating chromosome moves towards the dimmer, young centrosome.

cancer-stem cells. Even though gain/loss of non-disjoint chromosomes is only one of the causes of chromosomal instability in cancer cells, chromosome gain of non-disjoint chromosomes would not be random, but a favoured behaviour in cancer stem cells. Such a potential bias should thus be considered when modelling chromosomal instability in aneuploid cancer cell populations. Consistent with this hypothesis we note that the analysis of large human cancer samples reveals that whole chromosome gains (or losses) co-occur at much higher frequencies than combined chromosome gains and losses (*Ozery-Flato et al., 2011*). This phenomenon was so far thought to be the result of an evolutionary pressure; we propose that an asymmetric chromosome mis-segregation in cancer stem cells might provide a direct mechanistic explanation for this behaviour. Moreover, it could suggest a more general asymmetry in chromosomal instability, pointing to the need to determine whether other forms of chromosome mis-segregation, such as gain/loss of merotelic chromosomes, depend on centrosome age or not.

## Materials and methods

### Cell culture, drug, and siRNA treatments

HeLa, hTert-RPE1, and hTert-RPE1-eGFP-centrin1 cells (kind gift of A. Khodjakov) were grown in Dulbecco's modified medium (DMEM) supplemented with 10% FCS, 100 U/ml penicillin, 100 mg/ml streptomycin, at 37°C with CO$_2$ in a humidified incubator. HeLa-eGFP-centrin1 cells (kind gift of S. Doxsey) were further maintained in 500 µg/ml G418. HeLa-eGFP-centrin1/mCherry-CENP-A cells were generated by stably transfecting eGFP-centrin1 in HeLa-mCherry-CENP-A cells (kind gift of A. McAinsh, U. of Warwick); as HeLa-eGFP-centrin1/CENP-A-GFP, they were further supplemented with 500 µg/ml puromycin and 500 µg/ml G418. Live-cell imaging experiments were performed at 37°C in Lab-Tek II (Thermofisher, Switzerland) and Ibidi IV (Ibidi, Switzerland) chambers in L-15 medium supplemented with 10% FCS. To enrich for unaligned chromosomes, mitotic cells were removed by shake-off and the remaining cells treated with 10 ng/ml nocodazole for 2 hr. For nocodazole and monastrol washout experiments, cells were treated with either 1 µg/ml nocodazole or 100 nM monastrol for 4 hr (Sigma, Switzerland), washed twice with fresh medium and left to recover for 1 hr. Aurora-A was inhibited for 2 hr with 100 nM MLN8237 (Selleckchem.com, Switzerland), Plk1 for 2 hr with 100 nM BI2536 (Axon Lab AG, Switzerland), and CENP-E for 2 hr with 5 nM GSK-923295 (Chem Express, Switzerland). To stabilize microtubules, cells were treated with 5 nM Taxol (Sigma, Switzerland) for 2hr. To monitor anaphase cells, cells were released from a monastrol arrest and followed by live cell imaging. The following SiRNA oligonucleotides (Invitrogen and Thermofisher,

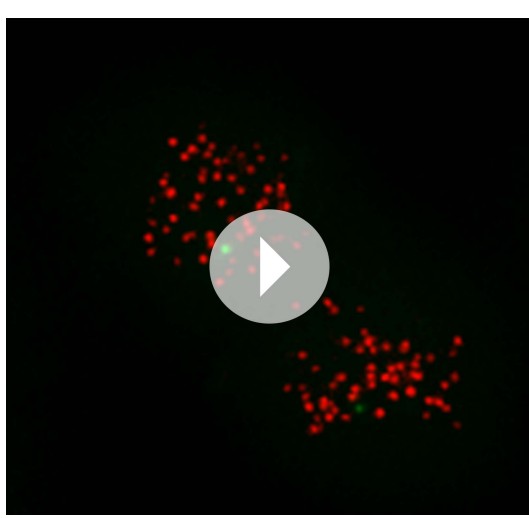

**Video 3.** HeLa cell expressing mCherry-CENP-A (kinetochore marker in red) and eGFP-centrin1 (centrosome age marker in green) depleted of Cenexin, after a monastrol washout. Note that the mis-segregating chromosome moves towards the dimmer, young centrosome.

Switzerland) were used: *siControl* (scrambled) (*Mchedlishvili et al., 2012*), *siNinein* (*Logarinho et al., 2012*), *siDsn1* (*Kline et al., 2006*), *siNnf1* (*McAinsh et al., 2006*), *siMCAK* (*Ganem and Compton, 2004*), *siCenexin* (OnTarget smart pool, L-017319-01-0005, Thermofisher); they were transfected using RNAi Max Lipofectamine (Invitrogen) and validated by immunofluorescence microscopy (*Figure 3—figure supplement 4*).

## Quantitative immunofluorescence

Cells were fixed with methanol at −20°C for 6 min, or with 20 nM Pipes (pH 6.8), 10 mM EGTA, 1 mM MgCl$_2$, 0.2% Triton X-100, 4% formaldehyde for 7 min at room temperature. For the microtubule nucleation assay, cells were incubated on ice for 1 hr before release in 37°C medium for 15 s for RPE-eGFP-centrin1 and 30 s for HeLa-eGFP-centrin1 cells. To image the attachment state of unaligned kinetochores, cells were rinsed with cytoskeleton buffer (10 mM MES, 150 mM NaCl, 5 mM MgCl$_2$, 5 mM glucose) prior and after fixation with 3% formaldehyde, 0.1% Triton X-100, and 0.05% glutaraldehyde for 10 min at room temperature. To analyze tubulin acetylation and detyrosination, cells were fixed with 20 nM Pipes (pH 6.8), 10 mM EGTA, 1 mM MgCl$_2$, 0.2% Triton X-100, 4% formaldehyde for 7 min at room temperature. To image the Calcium stability of kinetochore–microtubules, cells were treated with 0.5 nM CaCl2 dissolved in warm DMEM for 10 min at room temperature, rinsed with cytoskeleton buffer (10 mM MES, 150 mM NaCl, 5 mM MgCl$_2$, 5 mM glucose) prior and after fixation with 3% formaldehyde, 0.1% Triton X-100 and 0.05% glutaraldehyde for 10 min at room temperature. Three-dimensional image stacks of mitotic cells were acquired in 0.1- or 0.2-µm steps using 100x and 60x NA 1.4 objectives on an Olympus DeltaVision microscope (GE Healthcare, Switzerland) equipped with DAPI/FITC/TRITC/CY5 filter set (Chroma, Bellow Falls, VT) and a CoolSNAP HQ camera (Roper Scientific, Tuscon USA). 3D image stacks were deconvolved with SoftWorx (GE Healthcare) and analyzed with SoftWorx, Imaris (Bitplane, Switzerland) or ImageJ. For the nucleation assay, deconvolved total projections were analyzed as shown in *Figure 1B*. For the attachment status of unaligned kinetochores, single kinetochores were analyzed in 3D reconstruction of several z-stacks, and classified as shown in *Figure 3C*. To analyze Calcium stability, individual kinetochores were displayed in single z-planes and classified as shown in *Figure 3E*. The difference in microtubule nucleation capacity at old and young centrosomes was calculated as shown in *Figure 2—figure supplement 2*. The differences in protein levels at centrosomes or unaligned kinetochores at old and young centrosomes were calculated as shown in *Figure 3—figure supplement 1*. Images were mounted as figures using Adobe Illustrator. Primary antibodies used were mouse anti-CENP-A (1:2000, Abcam, United Kingdom), mouse anti-α-tubulin (1:10′000, Sigma), mouse anti-acetylated tubulin (1:1000; Sigma), rabbit anti-detyrosinated tubulin (1:1000, Merck-Millipore, Switzerland), rabbit anti-α-tubulin (1:500, Abcam), human CREST (1:400, Antibodies Inc, Davis USA), rabbit anti-phosphoT288-Aurora-A (1:1000, Cell Signalling, Danvers USA), rabbit anti-Plk1 (1:1000, Abcam), rabbit anti-Cenexin (1:1000, Abcam), rabbit anti-Ninein (1:500, Abcam), rabbit anti-Nnf1 (1:1000, *McAinsh et al., 2006*), rabbit anti-Dsn1 (1:2000, kind gift of Iain Cheeseman *Kline et al., 2006*), rabbit anti-MCAK (1:1000, *Amaro et al., 2010*), and rabbit anti-CENP-E (1:1000, *Meraldi et al., 2004*). Cross-adsorbed secondary antibodies were used (Invitrogen).

## Live imaging

To visually monitor the fate of non-disjoint chromosomes, Hela-eGFP-centrin1/mCherry-CENP-A cells were recorded every 2 or 4 min in 26 × 0.7-µm steps using a 60 × 1.4 NA objective on an Olympus DeltaVision microscope equipped with a GFP/mRFP filter set (Chroma) and a CoolSNAP HQ camera. To distinguish eGFP-centrin1 intensities in both experiments, reference images were taken at the end of the experiment, as three-dimensional stacks of 40 × 0.3-µm steps and a high-exposure times using the same objective and camera. Time-lapse movies were visualized in Imaris (Bitplane). To calculate distances, three-dimensional positions of the old and the young centriole and of all the kinetochores were detected using Imaris (Bitplane) and distances calculated with a custom MatLab function (see *source code 1*).

## Statistical methods

To check for biased distribution of polar chromosomes, Binomial probability test with expected probability success on a single trial of 0.5 was used. To calculate 2-ANOVAs, medians and median

confidence intervals, and run t-tests PRISM (GraphPad, La Jolla, CA) were used. Graphs were plotted in Excel (Microsoft, Redmond, WA) and PRISM and mounted in Adobe Illustrator (Adobe, Mountain View, CA).

## Acknowledgements

We thank the LMC of the ETH Zurich and the Bioimaging Core Facility of the University of Geneva for microscopy support, A Khodjakov for hTertRPE1-EGFP-centrin1 cells (Wadsworth Centre, USA), I. Cheeseman for Dsn1 antibodies (MIT, USA), A McAinsh for HeLa mCherry-CENP-A cells (Univ. of Warwick, UK) and S Doxsey (UMass, USA) for HeLa-eGFP-centrin1 cells. We thank A McAinsh and J. Meadows (both Univ. of Warwick, UK), M Gotta (Univ. of Geneva, Switzerland), H Maiato (IBMC, Portugal), V Panse (ETHZ, Switzerland) and the Meraldi lab members for critical discussions, and CH Tan and F Heydenreich for their contribution to *Figure 1B*. PM was funded by an SNF-project grant, the ETH Zurich, the University of Geneva and the Louis-Jeantet Foundation, IG by a Böhringer Ingelheim fellowship. IG and PN are part of the MLS PhD School.

The project was initiated and conceived by IG and PM, and directed by PM All experiments and data analysis were carried out by IG PN. helped IG. in the initial analysis of chromosome alignment in human cells. IG and PM interpreted the data and wrote the manuscript.

## Additional information

### Funding

| Funder | Grant reference | Author |
| --- | --- | --- |
| Schweizerische Nationalfonds zur Förderung der Wissenschaftlichen Forschung | 31003A_141256/1 | Patrick Meraldi |
| Université de Genève | | Patrick Meraldi |
| Louis-Jeantet Foundation | | Patrick Meraldi |
| Eidgenössische Technische Hochschule Zürich | | Patrick Meraldi |
| Boehringer Ingelheim Fonds | | Ivana Gasic |

The funders had no role in study design, data collection and interpretation, or the decision to submit the work for publication.

### Author contributions

IG, Conception and design, Acquisition of data, Analysis and interpretation of data, Drafting or revising the article; PN, Acquisition of data, Analysis and interpretation of data; PM, Conception and design, Analysis and interpretation of data, Drafting or revising the article

## Additional files

### Supplementary file

• Source code 1. Source code to calculate the distance between kinetochores and centrosomes at nuclear envelope breakdown

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
