## [Decision Letter]

Thank you for submitting your work entitled “Centrosome age regulates kinetochore microtubule stability and biases chromosome mis-segregation” for peer review at *eLife*. Your submission has been favorably evaluated by Tony Hunter (Senior Editor) and three reviewers, one of whom is a member of our Board of Reviewing Editors.

The reviewers have discussed the reviews with one another and the Reviewing Editor has drafted this decision to help you prepare a revised submission.

In this study, the authors test if centrosome age influences symmetric cell divisions using live and fixed cell analysis. The authors observed that unaligned chromosomes preferentially associate with old centrosomes with or without the nocodazole treatment to enhance the numbers of unaligned chromosomes. Using calcium treatment or RNAi depletion of proteins to perturb kinetochore microtubule stability, they go on to suggest that this bias occurs because kinetochore microtubule attachments are more stable at old centrosomes. Further, they demonstrate that the stability of these attachments also relies on the old centrosome protein cenexin. Finally, using live and fixed cell imaging, the authors show that this asymmetry contributes to a bias during chromosome non-disjunction with the daughter cell inheriting the old centrosome preferentially retaining the non-disjoined chromosomes. The observation that centrosome age influences chromosome alignment and imparts a functional asymmetry on mitotic spindles to influence the orientation of chromosome non-disjunction is quite novel and is a very interesting phenomenon.

Essential revisions:

1) The most important point: much stronger experimental support for the idea that centrosome age contributes to differences in kinetochore microtubule stability must be provided. One possibility would be to use microtubule photoactivation experiments to measure kinetochore microtubule stability in combination with a fluorescent marker of centrosme age, such as centrin 1. Perhaps detailed quantitative analysis of tubulin post-translational modifications could also be of some help here.

2) The underlying mechanism is mysterious - how do centrosomal proteins affect kinetochore fiber stability? Ideally, some additional experimental data on this point should be added. At the very least, a much better discussion of the possible mechanisms should be included.

3) In a broad sense, the data suggest that end-on kinetochore microtubule attachments precede sidewall microtubule attachments. But, that is counter to all other current models. How do the authors reconcile this discrepancy?

4) What is the difference between Figure 3 panels 1 and 2? Both are showing an end-on and unattached kinetochore. The cartoon diagram suggests that difference is the presence or absence of CENP-E but CENP-E localization is not determined in the experiment. The authors need to clarify their classification of attachment status. Also, the role of CENP-E is not clearly defined relative to microtubule dynamics, and this should be more clearly explained.

5) The authors present data suggesting that hyperstabilization by depletion of CENP-H abolishes the alignment bias but that “mild” stabilization by MCAK depletion does not. Can they provide a more quantitative assessment for the difference between CENP-H and MCAK depletion on microtubule stability (Figure 3)? It is unclear why MCAK depletion does not result in an unbiased distribution.

6) The authors state that in 2-5% of HeLa cells after monastrol wash-out they observe anaphases with non-disjoined chromosomes that preferentially segregate to the daughter cell with the old centrosome. However, the dominant defect reported by many labs for these types of washouts is lagging chromosomes (i.e. single chromatids) and not non-disjoined chromosomes. Do the authors observe lagging chromosomes and do these segregate preferentially to daughter cells with old centrosomes? Is the asymmetric segregation specific to non-disjoined chromosomes? How do the authors explain why the rate of non-disjoined chromosomes increases under these washout regimes?

7) It is not clear that the described phenomenon has a biological function. It might actually be just a consequence of the existing intrinsic centrosome asymmetry that cells have to deal with. This point deserves a better discussion.

8) The model figure needs revision. The authors claim that the two spindle halves have different kinetochore microtubule stability due to centrosome age but the diagram only shows the non-disjoined chromosome with different microtubule stability. Do they really mean that the difference in kinetochore microtubule stability is limited to just the lagging chromosomes? For this reason, it might be best to depict the half spindles differently.

9) The last paragraph of the Discussion randomly mentions cancer stem cells and chromosomal instability. This does not align with the rest of the text or data presented. There is nothing about cancer stem cells reported in the paper and chromosomal instability results from lagging chromosomes and not non-disjoined chromosomes. They authors need to revise the last paragraph.

---

## [Author Response]

1) The most important point: much stronger experimental support for the idea that centrosome age contributes to differences in kinetochore microtubule stability must be provided. One possibility would be to use microtubule photoactivation experiments to measure kinetochore microtubule stability in combination with a fluorescent marker of centrosme age, such as centrin 1. Perhaps detailed quantitative analysis of tubulin post-translational modifications could also be of some help here.

We agree with the reviewers that this point is crucial to the paper and that any experimental support for our hypothesis strengthens this study. The reviewers suggested using microtubule photoactivation to directly compare the turnover of kinetochore-microtubules at the two half-spindles. For this present study the photoactivation assay is, however, unsuited for two reasons. First, photoactivation is effective at quantifying large differences in kinetochore-microtubule stability, but the method often shows a high variability and is therefore unlikely to reveal small differences. For example, after the depletion of the plus-end binding protein APC and after photoactivation Bakhoum et al found an increase of 30% in kinetochore-microtubule stability compared to control depleted cells (4). This difference was, however, not significant, even though it is generally recognized that APC is an important microtubule plus-end regulator. This indicates that photoactivation cannot reproducibly reveal differences in k-fiber stability of 30% or less, which is more than the difference we would expect between the “old” and “young” half-spindle. The second caveat is specific to our particular experiment: the most sensitive tool to measure microtubule dynamics is a photoactivatable-GFP-α-tubulin, which would require a Cherry/RFP/Tomato-labeled version of centrin1. However, all our attempts to generate a red centrin1 that specifically labels centrioles in an age-dependent manner failed, as all constructs lead to protein aggregation of around the spindle poles (Alexey Khodjakov made very similar observations, and has never been able to produce a functional red Centrin1 construct). As an alternative, we have generated a cell line expressing the photo-convertible Eos-tubulin in conjunction with eGFP-centrin1; however the signal of Eos-tubulin masked the GFP-centrin1 signal at the poles, precluding any determination of centriole age.

To nevertheless test for a potential large difference in microtubule stability we used HeLa cells expressing PA-GFP-tubulin/H2B-RFP to “blindly” compare the dynamics at the two half-spindles, not knowing the age of the respective centrosome. We photoactivated GFP-tubulin along a one pixel wide line diagonally across the metaphase plate and monitored 3 photoactivated kinetochore-fibers in each spindle half. Our measurements revealed that the differences in dissipation of kinetochore-microtubules between the two half-spindles were as high as the differences amongst the k-fibers within a half-spindle, suggesting that any difference between the two half-spindles is likely to be small, and thus outside of the sensitivity range of the photoactivation assay.

As suggested by the reviewers we instead analyzed post-translational tubulin modifications as a read-out of kinetochore-microtubule stability. In particular, we focused on tubulin acetylation and detyrosination, two modifications that are associated with stable microtubules ([52]; Janke 2014). Whilst tubulin acetylation was easily detectable on k-fibers, tubulin, detyrosination was concentrated at spindle poles and diffusely present on the whole spindle (Figure 3—figure supplement 1). We therefore quantified tubulin acetylation at the plus-ends of k-fibers of individual sister-kinetochore pairs aligned on the metaphase plate, and the overall tubulin detyrosination in the two half-spindles (the heavily detyrosinated centrioles were excluded from our measurements). Our analysis revealed 20% more tubulin acetylation of kinetochore-fibers associated with the old centrosome. but no difference in tubulin detyrosination (Figure 3 and Figure 3—figure supplement 1). The strong difference in tubulin acetylation indicates that k-fibers ends associated to the old centrosome are more stable than the kinetochore-fibers associated with the young centrosome, fully supporting our hypothesis of a differential stability of k-fibers.

2) The underlying mechanism is mysterious – how do centrosomal proteins affect kinetochore fiber stability? Ideally, some additional experimental data on this point should be added. At the very least, a much better discussion of the possible mechanisms should be included.

We agree with the reviewers that the underlying mechanisms by which centrosomes affect kinetochore microtubule stability are mysterious and very interesting, but a better understanding of this phenomenon requires a thorough investigation that is beyond the scope of this revision. Nevertheless, as suggested by the reviewers we have added in the Discussion potential mechanisms by which centrosomal proteins could affect the dynamics of k-fiber plus-ends.

3) In a broad sense, the data suggest that end-on kinetochore microtubule attachments precede sidewall microtubule attachments. But, that is counter to all other current models. How do the authors reconcile this discrepancy?

We do not think that our data contradict the current models of how polar chromosomes are brought to the metaphase plate. There is a general consensus that kinetochores of chromosomes close to a spindle pole will first bind microtubules laterally, before dynein brings them closer to that same spindle pole. At this point the plus-end directed CENP-E will use lateral attachments to pre-existing k-fibers to move the unaligned chromosomes towards microtubule plus-ends to align them on the metaphase plate. Importantly, during the dynein-driven poleward motion, as they encounter a high density of microtubules, kinetochores may form a transient end-on attachment, resulting in monotelic or syntelic kinetochore-microtubule attachments. Monotelic and syntelic kinetochore-microtubule attachments are not frequent (less than 15% according to our data), as Aurora-B destabilizes these premature end-on attached kinetochores allowing them to revert to a lateral attachment, and thus favor CENP-E driven chromosome alignment onto the metaphase plate. Our data suggest that these transient end-on attachments close to spindle poles are more stable and therefore more persistent when kinetochores are bound to microtubules emanating from the old centrosomes; they remain however a transient state between lateral attachments, which is fully compatible with all existing models. To make these assumptions more explicit, we have now added 2 sentences paragraph 4 of the Results section.

*4) What is the difference between*
Figure 3
*panels 1 and 2? Both are showing an end-on and unattached kinetochore. The cartoon diagram suggests that difference is the presence or absence of CENP-E but CENP-E localization is not determined in the experiment. The authors need to clarify their classification of attachment status. Also, the role of CENP-E is not clearly defined relative to microtubule dynamics, and this should be more clearly explained.*

We apologize for the confusion: there is no difference between panels 1 and 2 in Figure 3, as in fact both contained one unattached and one end-on attached kinetochore. To improve the clarity of Figure 3 we have now removed panel 2, and point to the examples of end-on and unattached kinetochores in panel 1. To avoid further confusion, we have also removed CENP-E from the drawing, and rather illustrate how end-on attachments create a drag on CENP-E driven chromosome alignment in the final model in Figure 4. Indeed, our cartoon might have given the impression that the presence of CENP-E favors end-on attachment, which was not our message. Our model does not assume that CENP-E changes microtubule dynamics, but rather that it acts as a plus-end driven kinesin, as is widely recognized. We also have expanded the figure legend of Figure 3 to better explain our classification

*5) The authors present data suggesting that hyperstabilization by depletion of CENP-H abolishes the alignment bias but that “mild” stabilization by MCAK depletion does not. Can they provide a more quantitative assessment for the difference between CENP-H and MCAK depletion on microtubule stability (*Figure 3*)? It is unclear why MCAK depletion does not result in an unbiased distribution.*

With regard to the absence of an effect after MCAK depletion, we believe that it is consistent with the published data, as (4) has shown that MCAK depletion has no effect on microtubule dynamics in prometaphase, which would be the time when we observe unaligned chromosomes. To strengthen this interpretation we also performed a cold stable assay (Figure 3—figure supplement 3), which confirmed that in contrast to Dsn1 depletion, MCAK depletion had no major effect on kinetochore microtubule stability.

With regard to CENP-H depletion, after further analysis we realize that the situation is more difficult to interpret than we originally assumed. Indeed, our previous publication (2) showed that CENP-H depletion results in an ambivalent phenotype: a very dynamic plus-end that undergoes very rapid switches from microtubule polymerization to microtubule depolymerization, coupled to very stable k-fiber lattices. Even though CENP-H depletion reproducibly abolishes the bias in chromosome alignment, it is difficult to tell whether this is the result of the k-fiber stabilization or (more likely) the result of very dynamic plus-ends, which makes any interpretation hazardous. We therefore decided to remove the CENP-H treatment from this study and to replace it with low doses of taxol (5nM), as an unambiguous tool to stabilize k-fibers (Figure 3 and Figure 3—figure supplement 3). We confirm with a cold-stable assay that taxol stabilizes k-fibers, and we show that this stabilization abolishes the centrosome-age dependent bias in kinetochore-microtubule attachment and chromosome alignment. This perturbation confirms in a cleaner way that over- and de- stabilization of k-fibers is sufficient to erase the asymmetry induced by centrosome age.

6) The authors state that in 2-5% of HeLa cells after monastrol wash-out they observe anaphases with non-disjoined chromosomes that preferentially segregate to the daughter cell with the old centrosome. However, the dominant defect reported by many labs for these types of washouts is lagging chromosomes (i.e. single chromatids) and not non-disjoined chromosomes. Do the authors observe lagging chromosomes and do these segregate preferentially to daughter cells with old centrosomes? Is the asymmetric segregation specific to non-disjoined chromosomes? How do the authors explain why the rate of non-disjoined chromosomes increases under these washout regimes?

The main purpose of a monastrol-washout in the experiments shown in Figure 4 was to obtain a synchronous anaphase population, a fact that we now indicate more explicitly in the manuscript. As reported by many other laboratories after a monastrol washout in our eGFP-centrin1/mCherry-CENP-A cells we often saw single lagging kinetochores, which most likely reflect merotelic chromosomes. However, due to the fact that the mCherry-CENP-A signal is weak and that it tends to fade if one excites the fluorescence too often, we did not have sufficient temporal resolution to “track” the entire history of these single lagging kinetochores. Therefore we could not determine whether these single chromatids were segregated to the correct or incorrect daughter cell. This is now explained in the revised text. We agree that it would be interesting to know whether merotelically attached chromosomes also show a bias in chromosome mis-segregation, but since in most of the cases merotelics segregate normally (Cimini et al. 2004) and since such an investigation requires to know the precise history of those chromosomes, we believe that addressing this question will be challenging, as these are rare events, and that it would go beyond the time frame of this revision.

This is the reason why we focused only on those chromosomes where we were sure that both sister chromatids segregate to the same cell. As non-disjoined sister-chromatids remain attached to each other during anaphase, they move in a synchronous manner and their fate (correct segregation or mis-segregation) is easy to determine. We used monastrol washout as a way to synchronize cells, as such non-disjoined sister-chromatids are rare (2-5% of cells, with a chromosome mis-segregation event in 1-2% of the cells). Given the rarity of the event we have not been able to determine whether monastrol increases the frequency of such non-disjoined chromosomes, but so far we have not obtained any evidence for such an effect.

7) It is not clear that the described phenomenon has a biological function. It might actually be just a consequence of the existing intrinsic centrosome asymmetry that cells have to deal with. This point deserves a better discussion.

We agree, and have extended the Discussion to also mention the possibility that this asymmetry is related to a non-mitotic function.

8) The model figure needs revision. The authors claim that the two spindle halves have different kinetochore microtubule stability due to centrosome age but the diagram only shows the non-disjoined chromosome with different microtubule stability. Do they really mean that the difference in kinetochore microtubule stability is limited to just the lagging chromosomes? For this reason, it might be best to depict the half spindles differently.

We agree, and we have changed the model in Figure 4 to make all the microtubules emanating from the old centrosome more stable.

9) The last paragraph of the Discussion randomly mentions cancer stem cells and chromosomal instability. This does not align with the rest of the text or data presented. There is nothing about cancer stem cells reported in the paper and chromosomal instability results from lagging chromosomes and not non-disjoined chromosomes. They authors need to revise the last paragraph.

First, with regard to the origin of chromosomal instability (CIN), we agree with the reviewers that our last paragraph gave the unnecessary impression that CIN only arises due to non-disjoint chromosomes. However, at the same time we disagree with the reviewers that CIN exclusively results from lagging chromosomes, as non-disjoint chromosomes which can be frequent in cancer cells with replication defects, are clearly also able to lead to gain or loss of chromosomes (as visualized with our own experiments in Figure 4) or see (1). We therefore have revised our last paragraph to reflect that CIN has multiple origins, including non-disjoint chromosomes and lagging chromosomes.

Second, we also agree that our data are not directly related to cancer stem cells, and that our causal chain of events linking asymmetric gain and loss of chromosomes in human cancer tissues, with inheritance of old centrosomes in (cancer) stem cells, and our observed biased gain of chromosomes in cells inheriting the old centrosomes, is speculative. But we also would argue that the discussion part of a publication should/can also include speculative parts, to stimulate future investigations. We have thus taken this point in consideration and have emphasized the speculative nature of our proposed hypothesis, while believing that such a speculative part is not out of place.